# Role of Neoadjuvant Immune Checkpoint Inhibitors in Resectable Non-Small Cell Lung Cancer

**DOI:** 10.3390/ph16020233

**Published:** 2023-02-03

**Authors:** Ivy Riano, Inas Abuali, Aditya Sharma, Jewelia Durant, Konstantin H. Dragnev

**Affiliations:** 1Section of Medical Oncology, Dartmouth Cancer Center, Dartmouth Health, 1 Medical Center Drive, Lebanon, NH 03756, USA; 2Geisel School of Medicine, Dartmouth College, 1 Rope Ferry Road, Hanover, NH 03755, USA; 3Division of Hematology and Oncology, Massachusetts General Hospital, Harvard Medical School, 55 Fruit Street, Boston, MA 02114, USA; 4Department of Medicine, Dartmouth-Hitchcock Medical Center, Dartmouth Health, 1 Medical Drive, Lebanon, NH 03756, USA

**Keywords:** immune checkpoint inhibitors, non-small cell lung cancer, resectable lung cancer, neoadjuvant treatment, PD-L1, PD-1, CTL-4

## Abstract

The neoadjuvant use of immune checkpoint inhibitors (ICI) in resectable non-small cell lung cancer (NSCLC) is being increasingly adopted, but questions about the most appropriate applications remain. Although patients with resectable NSCLC are often treated with surgery and adjuvant chemotherapy or targeted therapies +/− radiotherapy, they still have a high risk of recurrence and death. In recent years, immune checkpoint inhibitors (ICI) (anti-PD-1/PD-L1 and anti-CTLA-4) have provided a new and effective therapeutic strategy for the treatment of advanced NSCLC. Therefore, it is possible that ICIs for early-stage NSCLC may follow the pattern established in metastatic disease. Currently, there are several ongoing trials to determine the efficacy in the neoadjuvant setting for patients with local or regional disease. To date, only nivolumab in combination with chemotherapy has been approved by the U.S. FDA in the preoperative setting, but data continue to evolve rapidly, and treatment guidelines need to be determined. In this article, we review the current preclinical and clinical evidence on neoadjuvant ICIs alone and combination in the treatment of early-stage NSCLC.

## 1. Introduction

Lung cancer is the leading cause of cancer mortality in the U.S. each year, representing nearly one-quarter of all cancer-related deaths [1]. Surgery is the cornerstone of curative treatment for early and locally advanced non-small cell lung cancer (NSCLC) [2]. About 20 to 25% of patients have resectable disease when diagnosed with NSCLC, yet it is estimated that 30 to 55% of patients who undergo curative surgery will eventually experience a recurrence of the disease, mainly at distant sites [3]. After surgery, the 5-year overall survival rate ranges from 92% in stage IA to 26% in stage IIIB [2]. Although neoadjuvant or adjuvant platinum-based chemotherapy can be used to treat certain stages [4], the improvement in 5-year recurrence-free survival and overall survival (OS), as compared with surgery alone, is only 5% [5].

In recent years, the advent of immunotherapy has greatly changed the landscape of thoracic oncology care. The use of immune checkpoint inhibitors (ICIs), compared with conventional therapeutic modalities, has been found to be safe and effective in patients with advanced NSCLC [6,7,8,9]. While adjuvant therapies for early-stage NSCLC have made significant advances [10,11], there is a continuing need for effective systemic treatments in perioperative settings. Currently, in patients with stage IB to IIIA epidermal growth factor receptor (EGFR) mutation-positive resected NSCLC, osimertinib demonstrated significantly longer disease-free survival when compared with placebo [11]. Similarly, the IMPOWER-010 trial showed a disease-free survival benefit with atezolizumab vs. placebo after adjuvant chemotherapy in patients with resected stage II–IIIA NSCLC, with more pronounced benefits in the subgroup whose tumors expressed PD-L1 on ≥1% of tumor cells [10], leading to the approval of atezolizumab in this setting by the U.S. Food and Drug Administration (FDA). Therefore, the neoadjuvant use of ICIs for resectable (stage I–IIIA) or potentially resectable (stage IIIB) lung cancer is currently a field of active, ongoing research. As a neoadjuvant treatment, immunotherapy may treat micrometastatic disease early and enhance the immune response while tumor antigens are still present [12,13]. This review aims to provide an overview of the role of ICIs in neoadjuvant settings in patients with resectable NSCLC.

## 2. Immune Checkpoint Inhibitors and Antitumor Immune Response

The process of cancer detection by the immune system features several key steps, known as the cancer-immunity cycle [14]. The first step is for cancer cells to release an antigen, which a dendritic cell will next encounter and later display on the cell surface with major histocompatibility complex (MHC) for presentation to a T cell. When T-cell receptors bind to the antigen/MHC complex, they require costimulatory binding of CD28 on the T cell and CD80/86 on the antigen-presenting cell for the T cell to become activated. Activated effector T cells will then travel through the vascular system until reaching the tumor site, at which point they will cross into the tumor bed and bind to the antigen target. With the binding to the target, effector T cells will proceed to kill the cancer cell, leading to the release of further cancer-specific antigens produced, thus continuing the cancer-immunity cycle [14].

One of the early requirements for a cancer cell’s survival and propagation is the task of avoiding host immune detection and destruction, described in the cancer-immunity cycle, and cancer cells possess many different mechanisms to accomplish this immune evasion [15]. One of the methods that cancer cells use to evade the immune system is the blockade of T-cell activation. This can be achieved through cytotoxic T-lymphocyte-associated protein 4 (CTLA-4), which, when present, will outcompete CD28 for the binding of CD80/86 [16] in a more dominant fashion, negatively regulating T-cell activation [17]. Specifically, CTLA-4 contains an immunoreceptor tyrosine-based inhibitory motif (ITIM) that transmits inhibitory signals in its cytoplasm. This effect weakens the immune response, contributing to the escape of tumor cells [18]. 

A second, incompletely understood, mechanism for blocking T-cell activation occurs through programmed cell death protein 1/programmed cell death ligand 1 (PD-1/PD-L1) binding [19]. PD-1 on T cells binds to PD-L1 on tumor cells, inhibiting T-cell-associated kinases and preventing cytotoxic T-cell responses. As a result, T cells cannot identify and destroy tumor cells, resulting in the failure of cytotoxic T lymphocytes and immune avoidance. Moreover, the binding of PD-1 to its ligand also inhibits T-lymphocyte proliferation, the production of cytokines, such as interleukin (IL)-2 and interferon (IFN)-Y, as well as the proliferation of B lymphocytes and the secretion of immunoglobulins, therefore reducing the effector cells’ immune effects. Immune escape is caused by a high level of PD-L1, as in NSCLC, where PD-L1 is expressed in 35–95% of the tumor cells (16). Moreover, there is often a correlation between high levels of PD-L1 expression in lung cancer tissues and high levels of T-cell infiltration, which often indicates that the antitumor properties of the T cells are depleted. The overexpression of PD-L1 within the tumor microenvironment is a mechanism that cancer cells use to protect themselves from antitumor immune response [20]. The expression levels of PD-L1 can predict treatment response to immune checkpoint blockade therapy that interfere in the PD-L1/PD-1 axis in cancer cells [20,21]. Although we have seen a rapid clinical translation of anti-PD/PD-L1 inhibitors, particularly in NSCLC [6,7,8,9], the regulation of PD-L1 expression on tumor cells remains poorly understood. The identification of these mechanisms of immune evasion prompted further preclinical investigations around the development of the inhibitors of CTLA-4 and PD-1/PD-L1, which are known as ICIs.

In 1996, it was discovered that the anti-CTLA-4 antibody (Ab) demonstrated potent antitumor effects with its administration in tumor-transplanted mice, leading to the rejection of the tumor, even when anti-CTLA4 was given a week after initial transplantation of the tumor [22]. Later, in 2002, the potential to target PD-1/PD-L1 as an ICI gathered sustained attention after a syngeneic mouse model demonstrated that administration of anti-PD-L1 Ab inhibited the growth of mouse myeloma cells, and that growth was completely blocked when myeloma cells were transplanted into PD-1 deficient mice [23]. These early studies served as the preclinical foundations and proof-of-concept for ICIs, which were necessary to move ICIs forward into the clinical setting. 

In recent years, preclinical investigations, vital for application in the clinical setting, have focused more on further understanding in depth the dynamic immune response mechanism of ICIs across organ systems and on exploring strategies to apply this understanding to maximize antitumor benefit [24]. Specifically, studies have been exploring the timing of ICI administration and examining the use of neoadjuvant vs. adjuvant immunotherapies in various cancers, including NSCLC. One of the earliest preclinical studies to examine adjuvant vs. neoadjuvant immunotherapy used a breast carcinoma mouse model to demonstrate that neoadjuvant ICI and surgical resection was able to eradicate metastatic disease, unlike what had been seen with surgical resection and adjuvant disease [24]. The improved antitumor benefit with the neoadjuvant administration of ICIs was also observed in other mouse models of NSCLC [25]. These studies demonstrate that the administration of ICIs in neoadjuvant settings may allow for the utilization of the primary tumor as an antigen source to develop a more robust antitumor immune response and longer-term immune memory of the tumor antigen for possible protection against future metastases [24,25,26].

### Effect of Neoadjuvant Immune Checkpoint Blockade on the Tumor Immune Microenvironment

A unique feature of preclinical immune checkpoint investigations in the neoadjuvant setting is the ability to study the in vivo effects of ICIs on the peripheral blood and tumor microenvironment, which is valuable in studying the details of the mechanisms of the antitumor effects seen with ICIs. A recent study in an orthotopic mouse model of early-stage KRAS-driven NSCLC found changes to the tumor microenvironment with treatments using anti-PD-1, which, with further investigation, may serve as useful predictive markers for disease response to treatment [27]. Prior to treatment with anti-PD-1 in this model, the tumor microenvironment was observed to support an immunosuppressive environment to promote tumor growth, with greater proportions of CD-4+ to CD-8+ T cells. Following treatment with anti-PD-1 in the early-stage model, the tumor microenvironment changed to favor an antitumor profile, with potent T-cell infiltration into tumor beds, greater proportions of CD-8+ to CD-4+ cells, increased CD-3+ T cells in tumor-burdened lungs, and an increase in polymorphonuclear myeloid-derived suppressor cells. Similar findings were also observed across clinical studies of the tumor microenvironment with neoadjuvant ICI treatment in NSCLC [28,29], supporting the validity of mouse models for use in tumor microenvironment investigations. Additionally, a preclinical study using an orthotopic 4T1.2 mouse model of spontaneously metastatic mammary cancer demonstrated that the effectiveness of neoadjuvant ICI treatment was dependent on functional Baft3⁺ and INF tumor changes, such as the initial step of CD-8+ T-cell recruitment into the microenvironment [30].

Moreover, studies have demonstrated profound changes in monocyte and macrophage subsets following anti-PD1/CTLA4 treatment, including elevated inducible nitric oxide synthase and decreased CD206 expression. Tumor-associated macrophages can respond actively to microenvironmental changes caused by checkpoint blockade, suggesting they should be considered when designing combinatorial strategies. Moreover, current data show that PD-L1 treatment can remodel the macrophage compartment, resulting in a more proinflammatory phenotype in both monocytes and macrophages. As a result of anti-PD-L1 treatment, macrophage polarization contributes significantly to enhancing T-cell responses, and that manipulation of macrophages in the tumor microenvironment can be used to augment the activity of anti-PD-L1 [31].

Characterizing the tumor microenvironment continues to be an exciting area of future study, about which relatively little is known. A more thorough understanding of the tumor microenvironment and its changes with treatment, such as neoadjuvant ICI, is expected to be valuable for the potential identification of biomarkers that could be used to inform more individualized treatment decisions in the clinic [32]. In the following sections, we discuss the clinical efficacy, safety, therapeutic strategies, challenges, and expectations of neoadjuvant ICIs in the treatment of resectable NSCLC.

## 3. An Overview of Neoadjuvant Immunotherapy Trials

### 3.1. Neoadjuvant Immunotherapy Alone

After immunotherapy showed survival benefits in metastatic NSCLC, investigations on its use in the neoadjuvant setting followed [33,34]. Prior research conducted by the NSCLC Meta-Analysis Collaborative Group on the utility of neoadjuvant chemotherapy in resectable NSCLC demonstrated a statistically significant benefit of preoperative chemotherapy on survival (hazard ratio 0.87, 95% confidence interval, 0.78–0.96, *p* = 0.007), and a 13% reduction in the relative risk of death (no evidence of a difference between trials), which only represented a modest improvement of 5% in absolute survival at 5 years [5]. However, since Forde et al. [35] reported positive data on the safety, tolerability, and major pathologic response (MPR) of PD-1 blockade therapy in the neoadjuvant setting of patients with resectable NSCLC, immunotherapy has rapidly transformed the landscape of neoadjuvant management for resectable NSCLC patients. In this pilot study (*n* = 21), two preoperative doses of the PD-1 inhibitor nivolumab were administered in adults with untreated, surgically resectable, early-stage (I, II, or IIIA) NSCLC, demonstrating no delays in surgery. An MPR occurred in 9 of 20 resected tumors (45%; 95% CI, 23–68), with 15% of all patients achieving a pathological complete response (pCR). The neoadjuvant nivolumab was associated with few side effects. Treatment-related adverse events of any grade occurred in 5 of 22 patients (23%; 95% CI, 7.8–45.4), and only one event was of grade 3 or higher. Furthermore, at the time of resection, tumors with an MPR had a higher frequency of T-cell clones that were shared between intratumoral and peripheral compartments, and a higher clonality of the T-cell population, as compared with tumors without an MPR. In early-stage lung cancer patients, achieving an MPR was highly associated with an increased tumor-mutational burden (TMB), similar to responses to PD-1 blockade in advanced NSCLC patients. These findings indicate that the TMB may be a primary determinant of the depth of pathological response to PD-1 blockade. Moreover, nivolumab also increased tumor infiltration of CD8+/PD-1-positive immune cells [35]. ICIs were found to be safe in the neoadjuvant setting; thus, this study paved the way for further clinical trials to be conducted to evaluate the efficacy of immune checkpoint blockade therapy in the preoperatory period.

Various clinical trials, including Keynote-223 (MK3475-223) [36], LCMC3 [37], and NEOSTAR [38], using an array of different primary endpoints, such as MPR, pCR, progression-free survival (PFS), and event-free survival (EFS), have demonstrated the benefit of perioperative immune checkpoint blockade in early-stage NSCLC. A summary of the most relevant clinical trials administering neoadjuvant immunotherapy in patients with early stages of NSCLC is shown in Table 1. 

Results of the phase-1 study Keynote-223 (MK3475-223) [36] demonstrated the neoadjuvant use of pembrolizumab as a safe approach in patients with stage-I and stage-II NSCLC. The study was a classic 3 + 3 design, where patients were divided into three dose-scheduled escalation cohorts based on the number of treatments rather than the dose: (i) a single pembrolizumab dose 3 weeks prior to surgery; (ii) and (iii) 2 pembrolizumab doses 3 weeks apart, with an interval from the last dose to surgery of (i) 3 weeks, (ii) 2 weeks, and (iii) 1 week. The MK3475-223 trial identified a tolerable safety profile as well as a positive MPR among 4 out of 10 total patients (40%; 95% CI, 16.7–68.8%), further demonstrating the potential benefit of immunotherapy use in the neoadjuvant setting. Interestingly, no correlation was seen between the levels of PD-L1 pretreatment and the pathologic response [36]. Similarly, the phase-2 PRINCEPS trial [39] used atezolizumab in patients with clinical stage-IA to stage-IIIA NSCLC (*n* = 30), without an MPR observed. However, the lack of response was attributed to the short delay between treatment with ICI and surgery, which occurred within 3–4 weeks after immunotherapy [39].

**Table 1 pharmaceuticals-16-00233-t001:** Most-relevant Published Clinical Trials of Neoadjuvant ICIs +/− Chemotherapy in Resectable NSCLC.

Clinical Trial	Phase	ICI	No. of Patients	MPR; pCR	≥Grade 3 Adverse Event
Immunotherapy Alone or Combinations					
Forde et al. (NCT02259621) [35]	Pilot	Nivolumab	21	45%; 15%	1%
Keynote-223 (MK3475-223) (NCT02938624) [36]	1	Pembrolizumab	10	40%; NR	NR
LCMC3 (NCT02927301) [37]	2	Atezolizumab	181	20%; 6%	11%
PRINCEPS (NCT02994576) [39]	2	Atezolizumab	30	Not observed	0%
IFCT-1601 IONESCO (NCT03030131) [40]	2	Durvalumab	46	NR *	0%
Gao et al. (ChiCTR-OIC-17013726) [41]		Sintilimab	40	40.5%; 8.1%	10%
NEOSTAR (NCT03158129) [38]	2	Nivolumab +/− Ipilimumab	44	22 vs. 38% 29 vs. 9%, respectively	13 vs. 10%
Reuss et al. (NCT02259621) [42]	1b/2	Nivolumab + Ipilimumab	9	NR; 33%	33%
NeoCOAST (NCT03794544) [43]	2	Durvalumab Durvalumab + oleclumab Durvalumab + monalizumab Durvalumab + danvatirsen	84	11.1%; 3.7% 19.0%; 9.5% 30.0%; 10.0% 31.3% 12.5%	0% 4.8% 0% 6.3%
Immunotherapy in Combination with Chemotherapy					
CheckMate 816 (NCT02998528) [44]	3	Nivolumab + platinum-based chemotherapy vs. platinum-based chemotherapy alone	773	NR **; 24 vs. 2.2%, respectively	33.5% vs. 36.9%, respectively
Shu et al. (NCT02716038) [45]	2	Atezolizumab + platinum-based chemotherapy	30	57%; NR	-
NADIM II (NCT03838159) [46]	2	Nivolumab + platinum-based chemotherapy vs. platinum-based chemotherapy alone	87	52% vs. 14% 36.2% vs. 6.8%, respectively	24% vs. 10%, respectively

ICI, immune checkpoint inhibitor; MPR, major pathological response; pCR, pathological complete response; NR, not reported. * Primary endpoint was complete surgical resection (R0) reported as 30% (*n* = 41). ** Primary endpoint was EFS reported as 31.6 months with nivolumab plus chemotherapy and 20.8 months with chemotherapy alone (HR 0.63; 97.38% CI, 0.43–0.91; *p* = 0.005).

The phase-2 LCMC3 [37] is the largest study to date to assess the safety and efficacy of ICI therapy in the neoadjuvant setting. Investigating the effect of preoperative atezolizumab monotherapy in 181 patients with stage-IB to stage-IIIB resectable NSCLC, it reported an MPR rate of 20% (*n* = 29; 95% CI, 14–28%) and a pCR rate of 6% (*n* = 8; 95% CI, 3–11%). Pathological response was significantly correlated with baseline PD-L1 tumor proportion score (TPS) (*n* = 111; *R*= −0.37; *p* < 0.001), with an MPR being achieved more frequently in patients with a tumor showing a TPS of ≥50. Similarly, an MPR in patients with TMB ≥ 16 mutations per Mb was 33% (5 of 15). Neither a radiographical response nor an MPR was observed in tumors with *EGFR* or *ALK* mutations. The safety profile was consistent with that observed in advanced disease. The most common treatment-related adverse events of grade ≥3 were pneumonitis (*n* = 4; 2%) and pneumonia (*n* = 3, 2%) [37].

With the growing body of evidence of a tolerable and safe profile and possible benefits in MPR and pCR, the phase-2 NEOSTAR trial [38] sought to further delve into the effect of single-agent (nivolumab) and combined-neoadjuvant immunotherapy (nivolumab/ipilimumab). Studying a total of 44 patients with resectable NSCLC, an MPR of 22% was demonstrated with nivolumab, while an MPR of 38% was demonstrated in the nivolumab/ipilimumab arm. The positive MPR findings with dual-ICI therapy were paralleled by a pCR rate of 29% with the combination vs. a rate of 9% with nivolumab monotherapy. No significant differences between the safety profiles were found; 13% demonstrated grade ≥3 immune-related adverse events in the nivolumab arm, and 10% demonstrated those in the dual-ICI group. A total of 89% patients who received at least one dose of a neoadjuvant ICI on trial had undergone successful resection. (A total of 39 patients underwent curative surgery, on or off trial). Using flow cytometry of samples obtained post-treatment, the NEOSTAR trial additionally revealed significantly higher levels of tumor-infiltrating lymphocytes (*p* = 0.057), non-effector tissue-resident memory T cells (*p* = 0.041), and effector tissue-resident memory T cells (*p* = 0.034) in samples treated with both nivolumab and ipilimumab vs. nivolumab monotherapy. These results suggest that the combination of immunotherapy led to higher levels of tumor-infiltration of both cytotoxic and memory cells, which may explain the higher levels of MPRs and pCRs seen in this study [38].

Similarly, Reuss et al. [42] conducted a multicenter, open-label, single-arm phase-1b/2 study with nivolumab plus ipilimumab in patients with resectable stage-IB to stage-IIIA NSCLC. However, 6 of 9 patients (67%) reported treatment-related adverse events, of which one-third of reported grade ≥3 toxicities (33%), leading to the early termination of the study [42]. The single-arm, phase-2 IFC-1601 IONESCO trial [40] was also stopped because of an excess of 90-day postoperative mortality (3 out of 4 deceased patients had cardiovascular comorbidities) after patients were treated with neoadjuvant durvalumab monotherapy. Yet, 41 patients (90%) achieved a complete resection (R0), and there was no grade ≥3 durvalumab-related adverse events [40]. MPRs and pCRs were not reported, and median OS and DFS were not reached. 

Nonetheless, the positive initial clinical studies on the neoadjuvant use of ICIs in resectable NSCLC have sparked further investigations into its use as single agent, as well as combined with other ICIs or novel agents (Table 2). The phase-2 NeoCOAST trial [43] used durvalumab alone (*n* = 26) and in combination with the anti-NKG2A antibody monalizumab (*n* = 20), the anti-CD-73 antibody oleclumab (*n* = 21), and the anti-STAT3 antisense oligonucleotide danvatirsen (*n* = 16) in the neoadjuvant setting to treat patients with stage-I—IIIA resectable NSCLC. These novel immunotherapy combinations resulted in higher MPR and pCR rates. The safety profile in the durvalumab monotherapy arm (treatment-related adverse events in 34.6% of patients) was similar to previously published data for anti–PD-1/anti–PD-L1 antibodies. No new safety signals were identified with any of the combination regimens [43]. Based on these results and the recent approval of neoadjuvant nivolumab plus chemotherapy, a follow-up, randomized clinical trial, NeoCOAST-2, was launched (NCT03794544). 

### 3.2. Neoadjuvant Immunotherapy in Combination with Chemotherapy

The concept of combining chemotherapy with immunotherapy was validated in metastatic NSCLC (Keynote 189, 407, IMPOWER-150). The addition of chemotherapy was hypothesized to lead to faster and deeper responses, with the release of more tumor antigens, thus augmenting the immune response and ICI activity. Efficacy in the neoadjuvant setting was evaluated in CHECKMATE-816 [44], a randomized, open-label trial in patients with stage-IB–IIIA resectable NSCLC. A total of 358 patients were randomized to receive either nivolumab plus platinum-doublet chemotherapy administered every 3 weeks for up to 3 cycles, or platinum-chemotherapy alone administered on the same schedule. The median EFS was 31.6 months (95% CI, 30.2, not reached) in the nivolumab-plus-chemotherapy arm and 20.8 months (95% CI, 14.0–26.7) for those receiving chemotherapy alone (HR 0.63 (97.38% CI, 0.43–0.91; *p* = 0.0052)). Among those treated with nivolumab plus chemotherapy, the pCR rate was 24%, and it was 2.2% (95% CI, 0.6–5.6) in those treated with chemotherapy alone. In the nivolumab-plus-chemotherapy group, 33.5% of patients experienced grade-3 or -4 adverse events, compared with 36.9% of those who received chemotherapy alone. There was a delay in surgery in 3.4% of patients receiving nivolumab plus chemotherapy, and in 5.1% of those receiving chemotherapy alone, leading to surgery cancellation in 1.1% and 0.6%, respectively. Moreover, in patients treated with nivolumab plus chemotherapy, the percentage of patients with ctDNA clearance was higher (56%; 95% CI, 40–71) than in the group of patients receiving chemotherapy alone (35%; 95% CI, 21–51). Patients with ctDNA clearance showed higher EFS and pCR rates than those without it, in both treatment groups [44]. On March 4, 2022, the U.S. Food and Drug Administration (FDA) approved nivolumab with platinum-based chemotherapy for patients with resectable NSCLC in the neoadjuvant setting. This represented the first FDA approval for neoadjuvant therapy for early-stage NSCLC [47].

Additionally, the phase-2, two-arm NADIM II trial [46] enrolled patients with resectable NSCLC to receive neoadjuvant nivolumab along with platinum-doublet chemotherapy for 3 cycles before surgical resection vs. chemotherapy alone. Of the 87 patients analyzed, neoadjuvant nivolumab plus chemotherapy significantly increased the pCR rate compared to chemotherapy in the ITT (36.2% vs. 6.8%; respectively, RR 5.25 (99% CI, 1.32–20.87); *p* = 0.0071). The combination of an immuno-chemotherapy regimen also improved MPR rates vs. chemotherapy alone (52% vs. 14%). Definitive surgery occurred for 91% of patients treated with adjuvant immunotherapy, and for 69% in the chemotherapy arm. According to data presented at the 2022 World Conference on Lung Cancer [48], patients who received the nivolumab and chemotherapy combination demonstrated 12- and 24-month PFS rates of 89.3% and 66.6%, respectively, compared with 60.7% and 42.3% for chemotherapy alone. Median PFS was not reached in the compared experimental cohort, and it was 18.3 months in the chemotherapy cohort (HR 0.48; 95% CI, 0.25–0.91; *p* = 0.025). The OS rates at 12 and 24 months were 98.2% and 84.7%, respectively, in the nivolumab-plus-chemotherapy group, compared with 82.1% and 63.5% in the chemotherapy group. Median OS was not reached in either arm (HR 0.40; 95% CI, 0.17–0.93; *p* = −0.034). Those results seem to reinforce the superiority of nivolumab in combination with chemotherapy vs. chemotherapy alone, significantly improving PFS, MPR and pCR rates in patients with stage-IIIA/B NSCLC.

### 3.3. Neoadjuvant and Adjuvant Immunotherapy Trials

With the U.S. FDA approval of the use of nivolumab in conjunction with chemotherapy as a neoadjuvant treatment and of atezolizumab as an adjuvant treatment of resected NSCLC, clinicians are faced with the clinical dilemma of how best to incorporate immunotherapy in the multimodality treatment of early-stage NSCLC. Several clinical trials have investigated the role of neoadjuvant and adjuvant immunotherapy, specifically trying to determine whether sequential neoadjuvant chemoimmunotherapy and adjuvant immunotherapy improve clinical outcomes. The TOP1501 trial (NCT02818920) [49] enrolled 30 patients with resectable NSCLC to receive two cycles of pembrolizumab, followed by surgery, and four cycles of adjuvant pembrolizumab; adjuvant chemotherapy was encouraged but not necessary. A total of 25 patients received surgical resection, with an 88% R0 resection rate, and one surgery delay. An MPR was observed in 28% of the patients, including 12% with a pCR. Pembrolizumab was safe and well-tolerated in the neoadjuvant setting, and there was no excess surgical morbidity or mortality. Data on survival outcomes are awaited.

The SAKK 16/00 trial (Swiss Group for Clinical Cancer Research) [50] administered 2 doses of neoadjuvant durvalumab after 3 cycles of platinum-based chemotherapy followed by surgery and adjuvant therapy with durvalumab alone for 1 year in patients with stage-IIIA NSCLC. Of the 55 patients who underwent resection, 34 (62%) achieved an MPR, and 10 (18%) had a pCR. Postoperative nodal downstaging (ypN0-1) was observed in 37 patients (67%). This study concluded that the addition of perioperative durvalumab to neoadjuvant chemotherapy in patients with stage-IIIA (N2) NSCLC is safe and exceeds the historical data on chemotherapy alone, with a high MPR and an encouraging 1-year EFS rate of 73%. Other trials that are currently underway are listed in Table 2. 

## 4. Use of Predictive Biomarkers

A major challenge in identifying patients who would most benefit from neoadjuvant ICI therapy is a lack of surrogate endpoints of clinical efficacy in many trials. While MPR has been shown to be a useful surrogate marker following neoadjuvant chemotherapy, its role following ICIs is less clear, with a significant number of patients with NSCLC found to be unresponsive to ICIs [51]. It is therefore imperative to identify predictive biomarkers that can assist in appropriate patient selection and to better guide clinical decision-making. 

The predictive role of PD-L1 expression has yet to be defined, constrained by heterogeneity among clinical trials utilizing different ICI agents in mono- and combination regimens, varying lengths of neoadjuvant therapies, and limited tissue availability in some cases. There are studies suggesting a positive correlation between high PD-L1 expression and an MPR [41,52], while others fail to demonstrate any association [45]. Analyses evaluating TMB have similarly conflicting results, and investigative efforts to determine utility are ongoing, especially since NSCLC tends to have a high TMB, as compared to other malignancies [35,53]. 

In the NADIM trial, TMB- and PD-L1-staining failed to show a correlation, but pretreatment circulating tumor DNA (ctDNA) was associated with survival [54]. ctDNAs are cell-free DNA molecules that are released into the bloodstream by apoptotic tumor cells, and they match the tumor’s somatic mutations. A potential predictor of prolonged survival in NSCLC patients treated with ICIs was noted to be the absence of, or reduced, ctDNA levels [55]. An emerging area of research investigates the feasibility of blood-based TMB (bTMB), which allows for the noninvasive and expedited testing that is reflective of ctDNA sequencing. This, however, may be of limited applicability in earlier stages of malignancy due to reduced tumor shedding, as compared to advanced, metastatic disease [56]. 

Other biomarkers under investigation include the composition of the gut microbiome and further analysis of the tumor-immune microenvironment, such as the intensity of CD8+ cell-infiltration and other immune-cell subsets [57]. Recent data suggest that integrating blood and tissue biomarkers, along with immune-PET (positron emission tomography) imaging may be a more accurate strategy for patient selection, allowing for a more risk-stratified approach [58].

## 5. Potential Pitfalls of Immunotherapy in the Neoadjuvant Setting

As the treatment paradigm shifts now towards the novel inclusion of ICIs in the neoadjuvant phase, new considerations emerge for the avoidance of potential pitfalls. First, it is imperative to ensure appropriate patient selection. Currently, the data available do not support attempts at “downstaging” patients with marginally resectable tumors via reliance on neoadjuvant ICIs +/− cytotoxic- and radiotherapy. The first step in clinical decision-making is the early involvement of thoracic surgery teams through multidisciplinary discussions to ensure that patients are appropriate surgical candidates with resectable disease [59].

Furthermore, there are insufficient data at present to determine the optimal ICI agent choice and duration. It is a thoughtful balance between administrations of a sufficient length of therapy to allow response while avoiding delays in curative surgical resections. This is further complicated by whether ICIs are administered alone, as a monotherapy, or in combination, and whether they are combined with chemotherapy and/or radiotherapy [60].

Concern arises about potential immune-related adverse events that can occur during neoadjuvant therapy and lead to delays in surgical treatment, albeit most were found to be manageable and did not prevent patients from undergoing surgery [61]. A recent systematic review found that treatment-related adverse events that were of grade 3 and higher were 0–20% in monotherapy ICI, 10–33% in dual-therapy ICI, 7% in chemoradiation–ICI, and 0–67% in chemo–ICI [62]. However, the management of immune-mediated side-effects, such as pneumonitis, can entail prolonged courses of systemic steroids, which can impact surgical timing and affect postoperative healing. In the same systematic review, it was also noted that the percentage of patients who failed to undergo resection in monotherapy ICI, dual-therapy ICI, chemoradiation–ICI, and chemo–ICI were 0–17%, 19%–33%, 8%, and 0–46%, respectively.

While the phenomena of pseudo- and hyper-progression with ICIs have been noted in the metastatic setting, and radiographic response to assessment can be inaccurate, it is overall felt that progression on neoadjuvant immunotherapy is much more commonly a typical disease progression and should be managed as such [38]. The thoughtful evaluation of imaging findings is warranted, however, with tissue biopsies needed in certain cases to differentiate true progression from pseudo-progression.

From a surgical standpoint, it is uncertain whether utilizing ICIs in the neoadjuvant setting can lead to intraoperative difficulties, such as perihilar fibrosis, which may be related to duration between systemic neoadjuvant therapy and timing of surgery [59]. It is also unclear whether pneumonectomies following neoadjuvant ICIs have a higher risk of complications, as data from the NEOSTAR trial showed a 6% incidence rate of bronchopleural fistulas and a 24% incidence of prolonged air leaks.

As the field moves into this new direction of combined ICI and cytotoxic- and radiotherapy in the neoadjuvant setting and survival outcomes mature, a focus on accurate pathologic evaluation and close assessment of treatment-related complications is essential to clearly understand therapeutic efficacy and develop a toxicity profile to help clarify future clinical guidelines.

## Figures and Tables

**Table 2 pharmaceuticals-16-00233-t002:** Ongoing clinical trials of neoadjuvant immunotherapy in resectable NSCLC.

Clinical Trial	Phase	ICI	Primary Endpoint
NeoCOAST-2 (NCT03794544)	2	Durvalumab + oleclumab or monalizumab or danvatirsen	MPR
NEOMUN (NCT03197467)	2	Neoadjuvant pembrolizumab	Safety, pathological response
AEGEAN (NCT03800134)	3	Neoadjuvant and adjuvant durvalumab + chemotherapy vs. chemotherapy alone	pCR, EFS
CheckMate-77T (NCT04025879)	3	Neoadjuvant nivolumab + chemotherapy followed by adjuvant nivolumab	EFS
KEYNOTE-671/MK-3475-671 (NCT03425643)	3	Neoadjuvant and adjuvant pembrolizumab + chemotherapy	EFS, OS
IMpower030 (NCT03456063)	3	Neoadjuvant atezolizumab + chemotherapy vs. chemotherapy alone followed by surgery, and open-label adjuvant atezolizumab vs. best supportive care	EFS
NCT03081689	2	Neoadjuvant nivolumab + chemotherapy	PFS
NCT05157542	1	Neoadjuvant durvalumab + chemotherapy + low-dose radiation therapy	Safety
NCT04379739	2	Neoadjuvant camrelizumab + antiangiogenic or chemotherapy	MPR
NCT04875585	2	Neoadjuvant pembrolizumab + lenvatinib	MPR
NCT04245514	2	Neoadjuvant durvalumab + chemotherapy + 3 cohorts of radiation	EFS
NeoDIANA (NCT04512430)	2	Neoadjuvant atezolizumab + bevacizumab + chemotherapy	MPR
NCT04699721	1	Neoadjuvant nivolumab + chemotherapy + probiotics	Safety
NCT05577702	2	Monotherapy neoadjuvant tislelizumab	MPR
NeoTAP01 (NCT04304248)	2	Neoadjuvant toripalimab + chemotherapy	MPR
NEOpredict (NCT04205552)	2	Neoadjuvant nivolumab + relatlimab	Feasibility
NCT03237377	2	Neoadjuvant durvalumab + tremelimumab + radiation	Safety
NCT05319574	2	Neoadjuvant tislelizumab + radiation	MPR
NCT04506242	2	Neoadjuvant camrelizumab + apatinib	MPR
NCT03871153	2	Neoadjuvant durvalumab + chemotherapy + radiation	pCR
NCT04638582	2	Neoadjuvant pembrolizumab + chemotherapy	ctDNA resolution
NCT04326153	2	Sintilimab	DFS

ICI, immune checkpoint inhibitor; MPR, major pathological response; pCR, pathological complete response; EFS, event-free survival; OS, overall survival; DFS, disease-free survival. Clinical trials currently registered at NIH ClinicalTrials.gov.

## Data Availability

Data sharing not applicable.

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
