# Peer review of "Role of Neoadjuvant Immune Checkpoint Inhibitors in Resectable Non-Small Cell Lung Cancer"

_pharmaceuticals, 2023, doi:10.3390/ph16020233_

Round 1
Reviewer 1 Report
Overall, a well-written review on neoadjuvant immunotherapy-based treatments in NSCLC, including all relevant RCTs.
Scientific interest is limited because several similar reviews are already available, with no added value emerging in this one.
additional comments:
1. What is the main question addressed by the research?
The authors provided a well-written review on neoadjuvant immunotherapy-based treatments in NSCLC, including all relevant RCTs.
2. Do you consider the topic original or relevant in the field? Does it address a specific gap in the field?
This is a relevant and emerging topic in the field. Still, several similar reviews are already available.
3. What does it add to the subject area compared with other published material?
The authors reviewed all relevant clinical trials providing a comprehensive overview of the topic. Further similar reviews already did the same and no specific added value is provided in this one. Some unclear themes in the field that could be better addressed in order to increase the scientific interest of this review are: the predictive role of oncogenic driver mutations, the timing of surgery, future directions…). Anyway this is a choice of the authors.
4. What specific improvements should the authors consider regarding the methodology? What further controls should be considered?
The methodology is correct in my opinion, except for Table 2 were only a few ongoing clinical trials are presented.
5. Are the conclusions consistent with the evidence and arguments presented and do they address the main question posed?
Yes
6. Are the references appropriate?
Yes
7. Please include any additional comments on the tables and figures.
No figures are provided.
Table 1: results in NeoCoast are not clearly understandable. It could be better to separate any type of drug combination
Author Response
Reviewer #1:
Comment 1: 1. What is the main question addressed by the research?
The authors provided a well-written review on neoadjuvant immunotherapy-based treatments in NSCLC, including all relevant RCTs.
Response 1: Thank you very much for reviewing the paper and for this comment.
Comment 2: 2. Do you consider the topic original or relevant in the field? Does it address a specific gap in the field?
This is a relevant and emerging topic in the field. Still, several similar reviews are already available.
Response 2: Thank you for your comment.
Comment 3: 3. What does it add to the subject area compared with other published material?
The authors reviewed all relevant clinical trials providing a comprehensive overview of the topic. Further similar reviews already did the same and no specific added value is provided in this one. Some unclear themes in the field that could be better addressed in order to increase the scientific interest of this review are: the predictive role of oncogenic driver mutations, the timing of surgery, future directions…). Anyway this is a choice of the authors.
Response 3: Thank you very much for your comment. Certainly, adding other topics such as the timing of the surgery would be interesting, however we have decided to leave those topics out of the scope of this review considering that the authors have experience mostly on medical oncology approach rather than surgical outcomes.
Comment 4: 4. What specific improvements should the authors consider regarding the methodology? What further controls should be considered?
The methodology is correct in my opinion, except for Table 2 were only a few ongoing clinical trials are presented.
Response 4: Thank you very much for your constructive comment. We have added 14 more ongoing clinical trials currently registered at the NIH ClinicalTrials.gov. Please refer to the update Table 2.
Comment 5: 5. Are the conclusions consistent with the evidence and arguments presented and do they address the main question posed?
Yes
Response 5: Thank you very much for your answer.
Comment 6: 6. Are the references appropriate?
Yes.
Response 6: Thank you very much for your answer.
Comment 7: 7. Please include any additional comments on the tables and figures.
No figures are provided.
Table 1: results in NeoCoast are not clearly understandable. It could be better to separate any type of drug combination
Response 7: Thank you for your suggestion. We agree with your observation, we have modified the table and separated the results by type of drug and combinations (please review table 1).

Reviewer 2 Report
The presented articles summarized current progress and potential challenges in neoadjuvant immune checkpoint inhibition therapy, specifically in the treatment of resectable non-small cell lung cancer. In my opinion, it is a comprehensive, and up-to-date review of relevant literature. The authors globally summarized published clinical trials in the discussed field. I would suggest acceptance after minor revision. Here are my comments/suggestions:
1. In section 2.1, could the authors include more details of the impact of neoadjuvant immune checkpoint blockage on tumor microenvironment? Such as the derivation of monocytes, activation of macrophage and dendritic cells as there are the major antigen-presenting cells to recruit T cells in the tumor microenvironment.
2. I strongly recommend the authors to include one or two figures illustrating the mechanism of neoadjuvant immune checkpoint blockage to improve the readability of this manuscript.
3. For the proposed pitfalls of this therapy, is there further discussion of potential strategies to overcome them?
Author Response
Reviewer #2:
Comment 1: The presented articles summarized current progress and potential challenges in neoadjuvant immune checkpoint inhibition therapy, specifically in the treatment of resectable non-small cell lung cancer. In my opinion, it is a comprehensive, and up-to-date review of relevant literature. The authors globally summarized published clinical trials in the discussed field. I would suggest acceptance after minor revision.
Response 1: Thank you very much for reviewing the paper and for this comment.
Comment 2: Here are my comments/suggestions:
- In section 2.1, could the authors include more details of the impact of neoadjuvant immune checkpoint blockage on tumor microenvironment? Such as the derivation of monocytes, activation of macrophage and dendritic cells as there are the major antigen-presenting cells to recruit T cells in the tumor microenvironment.
Response 2: Thank you very much for this observation. This is certainly a topic important to be included. We have added the following information in section 2.1.
“Studies have demonstrated profound changes in monocyte and macrophage subsets following anti-PD1/CTLA4 treatment, including elevated inducible nitric oxide synthase and decreased CD206 expression. Tumor-associated macrophages can respond actively to microenvironment changes caused by checkpoint blockade, suggesting they should be considered when designing combinatorial strategies. Moreover, current data show that PDL1 treatment can remodel the macrophage compartment, resulting in a more proinflammatory phenotype in both monocytes and macrophages. As a result of anti-PDL1 treatment, macrophage polarization contributes significantly to enhancing T-cell responses, and that manipulation of macrophages on the tumor microenvironment can be used to augment the activity of anti-PDL1.”
Reference: Huizhong Xiong, Stephanie Mittman, Ryan Rodriguez, Marina Moskalenko, Patricia Pacheco-Sanchez, Yagai Yang, Dorothee Nickles, Rafael Cubas; Anti–PD-L1 Treatment Results in Functional Remodeling of the Macrophage Compartment. Cancer Res 1 April 2019; 79 (7): 1493–1506.
Comment 3: 2. I strongly recommend the authors include one or two figures illustrating the mechanism of neoadjuvant immune checkpoint blockage to improve the readability of this manuscript.
Response 3: We appreciate your observation and agree that including a figure would be of great help, however, we do not want to incur Copyright issues. We will defer to including any figure in this manuscript. In addition, we have included more ongoing clinical trials in Table 2.
Comment 4: 3. For the proposed pitfalls of this therapy, is there further discussion of potential strategies to overcome them?
Response 4: Thank you for this constructive comment. We have commented in our last paragraph that further studies should focus on accurate pathologic evaluation and close assessment of treatment-related complications to clearly understand therapeutic efficacy and toxicity profile that could outline further treatment guidelines. Certainly, results from the ongoing clinical trials will help to elucidate potential strategies to shape further therapeutic interventions.
